# Risk Mapping of Influenza D Virus Occurrence in Ruminants and Swine in Togo Using a Spatial Multicriteria Decision Analysis Approach

**DOI:** 10.3390/v12020128

**Published:** 2020-01-21

**Authors:** Maxime Fusade-Boyer, Pidemnéwé S. Pato, Mathias Komlan, Koffi Dogno, Komla Batawui, Emilie Go-Maro, Pamela McKenzie, Claire Guinat, Aurélie Secula, Mathilde Paul, Richard J. Webby, Annelise Tran, Agnès Waret-Szkuta, Mariette F. Ducatez

**Affiliations:** 1IHAP, UMR1225, Université de Toulouse, INRA, ENVT, 31076 Toulouse, France; maxime.fusade-boyer@envt.fr (M.F.-B.); claire.guinat@envt.fr (C.G.); aurelie.secula@envt.fr (A.S.); mathilde.paul@envt.fr (M.P.); agnes.waretszkuta@envt.fr (A.W.-S.); 2Laboratoire Central Vétérinaire de Lomé, 55788 Lomé, Togo; patosteed@yahoo.fr (P.S.P.); fo_mathias@yahoo.fr (M.K.); koffipolo001@yahoo.fr (K.D.); dbatawui@yahoo.fr (K.B.); emilygomaro@yahoo.fr (E.G.-M.); 3St Jude Children’s Research Hospital, Memphis, TN 38105, USA; Pamela.McKenzie@stjude.org (P.M.); richard.webby@stjude.org (R.J.W.); 4Cirad, UMR ASTRE, 34398 Montpellier, France; annelise.tran@cirad.fr

**Keywords:** influenza D virus, Togo, risk mapping, spatial MCDA

## Abstract

Influenza D virus (IDV) has been identified in several continents, with serological evidence for the virus in Africa. In order to improve the sensitivity and cost–benefit of IDV surveillance in Togo, risk maps were drawn using a spatial multicriteria decision analysis (MCDA) and experts’ opinion to evaluate the relevance of sampling areas used so far. Areas at highest risk of IDV occurrence were the main cattle markets. The maps were evaluated with previous field surveillance data collected in Togo between 2017 and 2019: 1216 sera from cattle, small ruminants, and swine were screened for antibodies to IDV by hemagglutination inhibition (HI) assays. While further samples collections are needed to validate the maps, the risk maps resulting from the spatial MCDA approach generated here highlight several priority areas for IDV circulation assessment.

## 1. Introduction

In 2011, influenza D virus (IDV) was detected for the first time in pigs with influenza-like illness in the United States [1]. Since then, several studies have shown presence of the virus or anti-IDV antibodies in other species, including cattle, small ruminants, camelids, horses, and feral-swine [2,3,4,5]. The distribution of IDV is very wide since the virus has already been detected in America [6,7,8,9], Europe [10,11,12,13], Asia [14,15], and Africa [2], although the circulation of IDV in these continents, particularly Africa, is poorly understood. In Africa, serological evidence of the IDV circulation has been found in Benin, Togo, Morocco, and Kenya, but the virus has not yet been isolated on the continent [2].

Experimental infection with IDV is associated with mild to moderate pathogenicity in calves [8,16] and in the field, IDV is more often detected in cattle with respiratory clinical signs rather than without (OR = 2.94), although not significantly [17]. IDV has, thus, been postulated to play a role in the bovine respiratory complex [8,16,17,18].

Active surveillance of animal influenza A viruses has been carried out in Togo from 2008–2013 [19], and again since 2017 [20]. IDV circulation was first assessed in a preliminary study with samples from 2009 through 2015. Twenty one percent of the cattle sera tested in 2015 were seropositive for IDV [2]. Because of field, personnel, and economic limitations, surveillance activities are difficult to initiate and maintain in Togo and other countries in Africa. It is, therefore, important that any resources spent on such activities should be optimally utilized. In the present study, risk maps were drawn with the aim to implement a risk-based surveillance system of IDV in Togo. To increase the efficiency and sensitivity of IDV surveillance, risk maps of IDV occurrence in Togo were generated using a spatial multicriteria decision analysis (GIS-MCDA) and compared with available surveillance data. GIS-MCDA is based on existing knowledge of the disease but contrary to statistical models, it does not require reliable epidemiological disease data which are not available for IDV in Togo [21]. GIS-MCDA operates in several well-defined steps: risk factors identification and definition of a mathematical relationship between risk factors and disease occurrence, risk factors weighting, combination of risk factors, risk maps drawing, and finally, uncertainty analysis and map validation when possible. Spatial MCDA has been used in Africa to assess the likelihood of occurrence of Rift valley fever and African swine fever, and the likelihood of highly pathogenic avian influenza H5N1 virus introduction and spread [22,23,24].

## 2. Materials and Methods

### 2.1. Surveillance Data

Through the national disease surveillance program, 2412 samples were collected from cattle, small ruminants, and pigs, in markets and farms in 22 different locations in Togo between 2017 and 2019. These samples included 1196 nasal swabs and 1216 sera (not necessarily paired samples from the same individuals). Nasal swabs were pooled in groups of 5 and RNA extraction was carried out using the RNeasy® Mini Kit (Qiagen). RNAs were screened for IDV by RT-qPCR using One-step RT-PCR kit (Qiagen) with primers and TaqMan probe targeting PB1 gene, as described by Hause et al. [1]. All sera were treated with receptor destroying enzyme (RDE, Seika) following the manufacturer’s instructions, diluted 10 folds, and hemadsorbed on packed chicken red blood cells. Hemagglutination inhibition (HI) assays were performed as previously described [16], with four hemagglutination units of D/bovine/France/5920/2014 and 1% chicken red blood cells. Based on our previous study carried out on African sera, a single antigen was used in HI tests, although two distinct genetic and antigenic lineages of IDV have been reported so far in North America and Europe (D/swine/OK and D/bovine/660) [7] (no significant differences were observed between the 2 antigens in HI assays on African sera [2]). Sera with antibody titers ≥ 10 were considered positive. 

### 2.2. Identification of Risk Factors and Experts Survey

First, nine international IDV experts were selected based on their recognized IDV experience. As the nine international experts did not know the Togolese context, four local experts from the Togolese veterinary services and field veterinarians were also selected for their knowledge on the livestock system, cattle market management, and local wildlife. Nine risk factors associated with IDV were selected based on the literature and submitted to international experts for their evaluation (Table 1). 

In order to assign a weight for each risk factor and to determine the relationship between the risk of occurrence and risk factors, international IDV experts were asked to answer an online questionnaire developed using the koBo Toolbox (https://www.kobotoolbox.org/). The questionnaire was composed of three parts. First, experts were asked to select relevant risk factors among those that were previously identified (Table 1). Second, they were asked to characterize the relationship between each selected factor and the risk of IDV occurrence by selecting from a list of several mathematical functions (linear, sigmoidal, quadratic and linear bi-directional). For non-linear relationships, experts could specify thresholds. To make this step easier without influencing answers an example was used with malaria, which illustrated the relationship between mosquito density and malaria risk. Third, we used the analytical hierarchy process (AHP) to assign a weight to each risk factor [28]: experts were asked to fill in a pair-wise comparison matrix where each factor was compared with the others, relative to its importance, from 1/9 (“extremely less important”), through 1 (“equal importance”), to 9 (“extremely more important”). The relationships and thresholds were also discussed with the local experts who were often the best qualified individuals to define the relationships and the thresholds due to their direct field expertise. When it was not possible to specify a threshold, the default relationship selected was linear. Some risk factors were clearly relevant to sampling schemes, but they could not be represented on a map. Such risk factors were included in the expert opinion questionnaire, but then removed to generate the maps. The weights of the remaining risk factors used to make the map were corrected as described below, in order to keep risk values between 0 and 1.

### 2.3. Spatial Data Collection and Geoprocessing

The raster density maps of domestic animals (pigs, goats, sheep, and cattle) were obtained from the portal to spatial data and information Geonetwork from the FAO (http://www.fao.org/geonetwork/srv/en/main.search?any=Livestock+GLW&hitsPerPage=10). The rasters corresponding to sheep and goat densities were congregated to obtain a small ruminants density map. The vector map of water area distribution was obtained from DIVA-GIS (https://www.diva-gis.org/). The vector map of cattle markets places was generated with data from the ministry of agriculture of Togo and from a shapefile of cities loaded from DIVA-GIS. As we were unaware of any wildlife distribution data for Togo, a vector map showing distribution of the main forests and wildlife protected areas of Togo, extracted from OpenStreetmap of QGIS, was used as a proxy. The vector map of transhumance area was drawn thanks to data from the ministry of agriculture of Togo. In order to combine the different layers, all initial layers were geoprocessed as follows. First, all vector layers were transformed in raster files. Second, the layers’ values were modified to range between 0 and 1 by using the fuzzy functions corresponding to the relationships selected by the experts [29]. The layers where the risk is linked to proximity (cattle markets, water, wildlife, and transhumance areas), were geoprocessed before fuzzy transformation by using the Euclidean distance function in ArcGIS. While the distance to the outer border of water, wildlife, and transhumance areas were considered, cattle markets were treated as centroids. All layers were standardized with the same resolution: 0.0083 × 0.0083 decimal degree.

### 2.4. Generation of the Final Maps

Two maps were obtained for each expert, one corresponding to the transhumance period and a second one corresponding to the period outside the transhumance. For each international expert, a map was drawn by applying the weight previously calculated from the pairwise matrix to the corresponding layer. Because maps from the different experts were similar, a mean weight for each risk factor was calculated for both periods. This final weight was then used to generate the final maps applying the following equation to risk factors layers, where *n* is the number of risk factors, w*_i_* is the weight, and RF*i* is the value of risk factor *i.*
$$Suitability\;index=\;\overset{n}{\underset{i=1}{\sum }}w_{i^{*}}RF_{i},\;1\leq i\leq n$$

To generate the maps corresponding to the period outside the transhumance or those where some risk factors were removed as they could not be spatially represented, the relevant layers were removed from the model and the other weights were corrected using the following equation. In this equation, $w_{c_{{\mathrm{RF}}_{i}}}$ is the corrected weight for the risk factor *i*, $w_{RF_{i}}$ is the original weight previously calculated by the expert for the risk factor *i*, and $w_{RFex}$ corresponds to the weight of risk factors which were excluded with *n* as the number of excluded risk factor.
$$w_{c_{{\mathrm{RF}}_{i}}}=\frac{w_{RF_{i}}}{1-\sum_{j=1}^{n}w_{RFex}},\;1\leq j\leq n$$

### 2.5. Uncertainty Analysis and Validation

To assess the robustness of the model and determine the impact of weight variations on the final maps, an uncertainty analysis was carried out. For this step, all the layers corresponding to each risk factor were converted into shapefiles. The shapefiles were then merged to obtain a final shapefile with all the risk factors using the animal density shapefile for the spatial joining process as a reference in order to have the maximum spatial resolution. The dbf file from the final shapefile was processed in Rstudio for uncertainty analysis measuring the standard deviation of each point of the map when the weight of each risk factor varied from −25% to + 25% of the weights defined by the experts [30]. When the weight of a given risk factor was adjusted, the weight of the others was modified in order to keep a sum of all weights equal to 1. In total, 100 maps (during transhumance and outside the transhumance period) were generated and the standard deviation was mapped on the final shapefile.

### 2.6. Risk Maps Comparison with Serological Results

Risk maps were visually compared with serological results obtained from cattle, small ruminants, and swine. Among all the sera collected over a given area, if at least one was found seropositive for IDV, the area was considered as positive for the comparison with risk maps. 

## 3. Results 

### 3.1. Surveillance Data

All nasal swabs were found negative for IDV by RT-qPCR regardless of species sampled. A seropositivity rate of 4.5% was found in cattle and of 3.8% in small ruminants. All sera samples from swine were seronegative (Table 2). Surprisingly, 43 positive sera were collected during the transhumance period and only three were outside the transhumance period.

### 3.2. Risk Mapping 

Three of the nine international experts from three different continents answered consistently to all questions and their answers could be used for the present study. According to all experts, the most important risk factors of IDV occurrence were those directly linked to cattle; namely, cattle density, cattle markets, presence of respiratory clinical signs, and cattle age. A linear increasing relationship in animal densities was used, as no specific threshold was identified by the experts. Sigmoid decreasing relationships were used for the proximity with markets, wildlife, water, and transhumance areas, with the greatest risk between 0 km and threshold *a*, decreasing risk thereafter and negligible after threshold *b* (Table 3). Risk factors “cattle age” and “respiratory clinical signs in cattle” were ignored for the maps because they could not be spatially represented.

### 3.3. Suitability Map, Uncertainty Analysis, and Serological Comparison

The most suitable areas for IDV occurrence were those containing cattle markets (spots with highest risk values on Figure 1A) and areas with high cattle density (diffuse yellow areas on Figure 1A). 

During the transhumance period, areas where cattle from neighboring countries are kept showed an increased IDV occurrence risk (Figure 1B). Irrespective of the period, some areas seemed at higher risk than others, especially cattle markets located in the North-West of Togo. 

Regarding the uncertainty analysis, for both maps, the maximum value of standard deviation was far from the 0.1 value, supporting the robustness of the model (Figure 2). Important changes in the weights defined by experts had a very weak impact on the final model. Variations of more than 20% in the value of the weight previously defined by experts induced a maximum change of 0.0287 of the risk value. 

Because of the non-random sampling plan, it was not possible to correctly validate the maps using the available serological results. Nevertheless, a comparison of risk maps and serological results highlighted some high-risk areas of IDV occurrence, which had not been previously considered for sample collection (Figure 3). Encouragingly, visual comparison of risk maps and serological results showed that most of the sampled areas with no seropositivity were deemed at low risk using our model. Seropositive samples came from a mix of areas considered as high and low risk (Figure 3).

## 4. Discussion 

In the present study, risk factors and areas at higher risk of IDV occurrence were identified in Togo. Cattle markets and high-density areas seem at higher risk of IDV occurrence, and especially the cattle market in the North-West of Togo, which has not been sampled to date. According to uncertainty analysis, the North of Togo is the most variable region when changing weights, with a standard deviation value which remained inferior to 0.1. 

We were unable to detect IDV in any nasal swab collected during the study period, likely because only a limited number of nasal swabs had been collected in cattle, the main host of the virus. Alternatively, the negative results could be due to the short time window to detect the virus since IDV is shed for about 10 days in calves under experimental conditions [16]. Clearly, increasing sampling intensity and prioritizing young cattle with respiratory clinical signs is necessary; both these factors were strongly associated with IDV according to expert’s opinion. Seropositivity rates we calculated may also have been underestimated because HI assays are less sensitive than ELISAs, which should be preferred for further serological analyses [31], even if previous IDV seroprevalence in Africa were calculated from HI assays data as well [2].

As no IDV vaccination is in place, all the positive sera came from natural infections. The seropositivity rates observed in cattle and small ruminants were lower than those reported in Europe, North America, Asia, or Africa [2,3,12,13,15,26,27]. Our IDV seroprevalence estimated in Togo was also higher in 2015 than in 2017–2019, reaching 21% in cattle [2]. Interestingly, the HI titers in seropositive animals were also lower in the current samplings than in the previous years. It should also be noted that IDV has not yet been isolated in Africa, and the local strains may be antigenically distinct from the strain used for the HI assays (D/bovine/France/5920/2014) as was previously suggested [7]. The temporal differences in the seropositivity rates in cattle in Togo could also be explained by differences in locations of samples collection. In 2015, most samples were collected in the Adetikopé market, which receives cattle from throughout Togo and neighboring countries. In the present study, samples were collected in peri-urban farms in Lomé, areas where cattle have less contact with animals from other places. 

Interestingly, in the present study, 43 positive sera among 723 were collected during the transhumance period and only three positive sera among 493 were collected outside the transhumance period. These results should, however, be considered with caution. Indeed, all the 399 cattle sera were collected during the transhumance period, which represents a significant bias. Another significant bias is the collection areas, since some places were sampled only during transhumance period and some others were only sampled outside the transhumance period. According to the risk maps (Figure 1), the North of Togo (where the highest density of cattle is observed) is at higher risk of IDV occurrence; however, the region was poorly sampled during the collection period. Nevertheless, cattle markets are at risk and should be prioritized for future sampling campaigns. 

The MCDA approach has some inherent limits that should be taken into consideration. The method can only consider risk factors which can be mapped. To address this limitation, risk factors that could not be represented spatially were used in conjunction to the map, to further inform future sampling strategies. While no clear guideline on the number of experts to select for eliciting health topics can be found [32], only three IDV experts fully answered the questionnaire. This number is very limited, but it is in line with the literature [33], and the answers from the three experts were very similar. 

An additional caveat of the final maps is that they are influenced by the quality of the layer data used to generate them. As far as animal density is concerned, the raster layers available from the FAO were generated with data from 2010, 7 to 9 years prior to the study period. According to local experts, while animal density has increased for cattle, small ruminants, and pigs, the areas with the highest animal densities in Togo have not changed. All the density animal raster layers showed « no data » for two geographic locations corresponding to wildlife protected areas (Figure 2). According to local experts, the density of domestic species is low in these locations but probably not null. This should, however, not represent an important issue since the dominating risk factors in the two areas are proximity to water and proximity to wildlife, both risk factors with the lowest weights in the model. No pigs were recorded from the FAO dataset in the North of Togo, which was not representative of the true density according to the local experts (personal communications). The risk of occurrence of IDV in the North of Togo is thus probably underestimated because of these missing data.

Finally, in the risk factors, we chose to differentiate “proximity to water” and “cattle markets”, despite both being gathering risk factors. We rationalize this as they were given different levels of risk as underlined by the experts’ answers. Indeed, cattle in cattle markets were coming from throughout Togo and sometimes from other countries, whereas cattle and other species at water areas were coming from a much more limited geographic area. 

The validation of knowledge-driven models like MCDA is very challenging because of the absence of complete epidemiological data, which in itself is often a driving factor to why the MCDA approach is used in the first place. Despite this challenge, spatial MCDA has been validated in several studies in different countries and on different diseases (for example avian influenza in Asia and African swine fever and Rift valley fever in Africa), underlining the benefit of using this method for risk-based surveillance [22,23,34,35,36]. In our study, the visual comparison of risk maps with serological results from the field showed that most of the sampled areas with negative results were at low risk of IDV occurrence whereas the positive samples were in high and low risk areas of the maps. Positive serological results in low risk areas can be explained by the fact that ruminants are bred to an older age in Togo than in western production systems, resulting in a higher probability to identify a seropositive ruminant which could have been infected by IDV months or years earlier and possibly in another location. Because of the non-random approach used to collect the field samples, it was not possible to calculate the exact IDV seroprevalence for each sampled area, which is necessary to more fully validate the map. Some risk areas were not sampled, including the area with the highest risk (0.75) in the North-West of Togo. Moreover, no samples were collected in cattle, the main host of the virus, in cattle markets, and in the North of Togo, high risk areas. Nevertheless, the highest titers in cattle were observed in samples collected during the transhumance period at a transhumance area, which is consistent with transhumance as a risk factor for IDV occurrence. 

Clinical signs in cattle are not specific of IDV, and IDV surveillance represents a significant financial and human cost. Thus, any method that helps to optimize future surveillance is valuable in aiding the understanding of IDV circulation in Africa and its evolution. Despite several limits, spatial MCDA is rapid to implement and can be very useful to identify areas where surveillance should be focused. In this context, the use of risk maps is a powerful tool to maintain an efficient surveillance with a better-balanced cost–benefit ratio. 

## Figures and Tables

**Figure 1 viruses-12-00128-f001:**
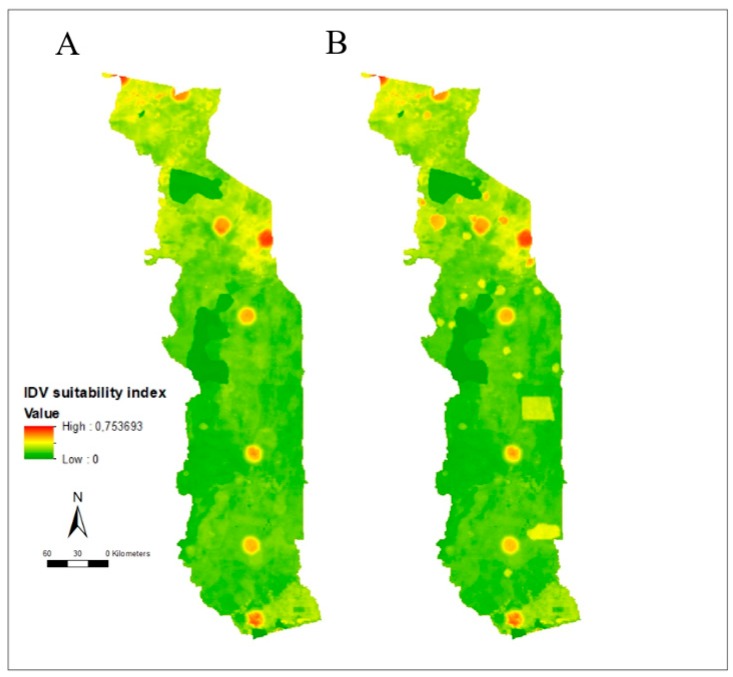
Suitability map for occurrence of influenza D virus in Togo. (**A**) Outside the transhumance period. (**B**) During the transhumance period.

**Figure 2 viruses-12-00128-f002:**
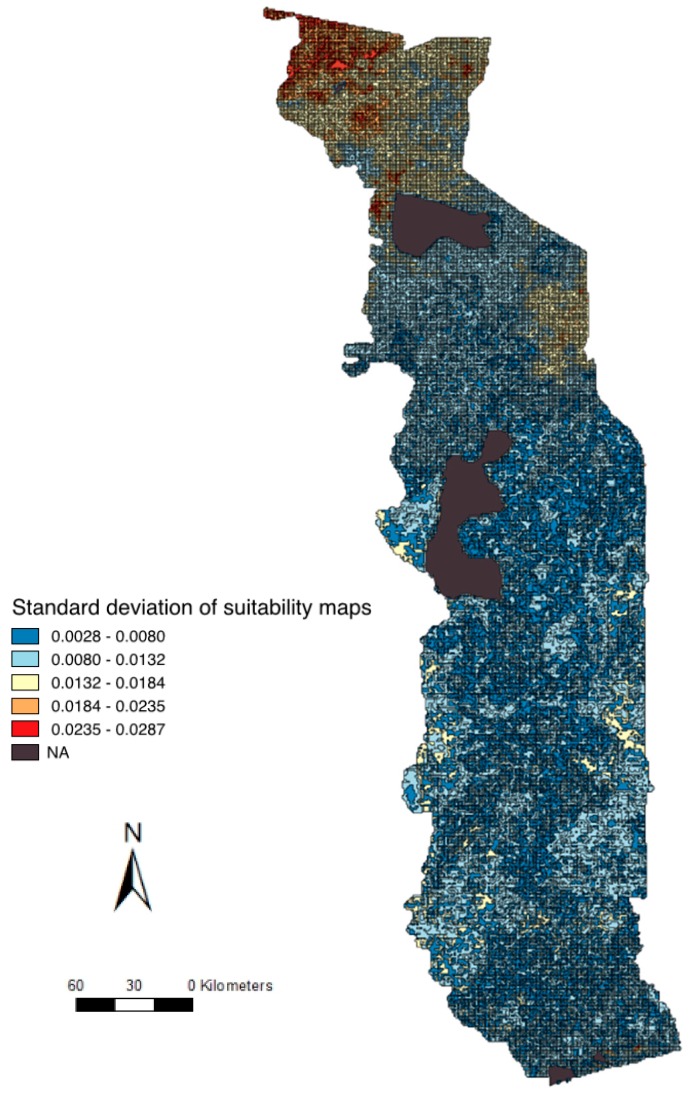
Uncertainty map: standard deviation of suitability maps for IDV occurrence outside the transhumance period.

**Figure 3 viruses-12-00128-f003:**
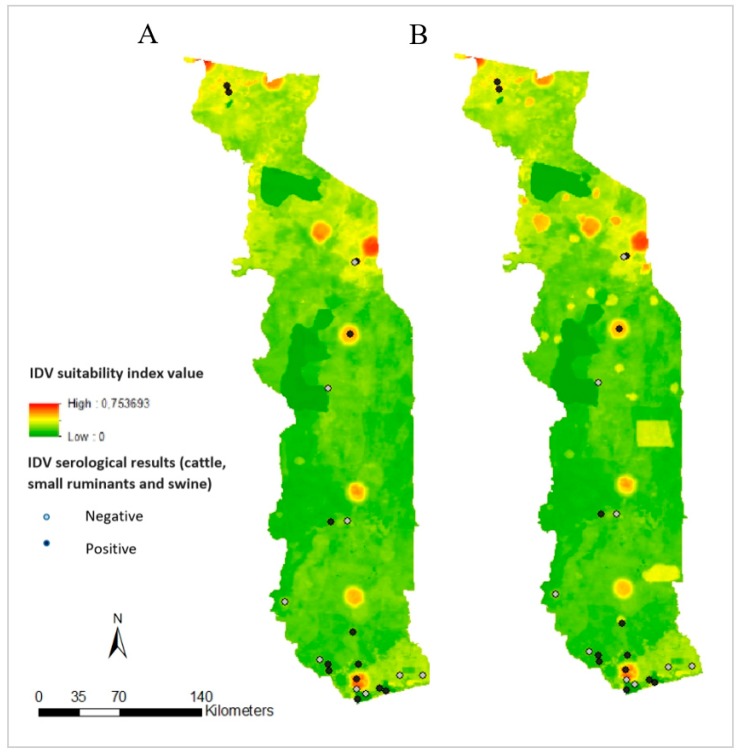
Risk maps comparison with serological results. (**A**) Outside the transhumance period. (**B**) During the transhumance period.

**Table 1 viruses-12-00128-t001:** Risk factors selected for risk mapping of IDV occurrence.

Risk Factor	Explanation	References
Swine density	IDV was discovered in swine and it is efficiently transmissible in this species.	[1,25]
Cattle density	Cattle are susceptible to IDV and harbor the highest seropositivity rates. Cattle are considered as the main host of the virus. IDV is also transmissible by aerosol between cattle.	[8,12,13,16,26,27]
Small ruminants density	Specific antibodies against IDV were detected in small ruminants, justifying their density as a risk factor.	[3,26]
Presence of respiratory clinical signs in cattle	Several studies report that IDV is more commonly isolated from cattle with respiratory clinical signs and can be airborne transmitted among cattle.	[8,16,17,18]
Cattle age	Calves appear more susceptible to IDV infection than adults.	[8,27]
Proximity to cattle market	Some cities in Togo receive cattle from all over the country and sometimes from neighboring countries. Cattle can stay in fields around the city waiting to be transferred to the slaughterhouse or to be sold to other farmers. Cattle markets represent focus points where cattle of different sanitary status and from different origins are parked, likely leading to an easier circulation of the virus.	Local expert opinion
Transhumance areas	Transhumance occurs each year in Togo between January and May. During this period, about 50,000 cattle come from Sahelian countries and are parked on dedicated fields, with the possibility of contact with local cattle. Trade with local farmers occurs during this period. Transhumance areas and periods were therefore considered a risk factor for IDV occurrence.	Local expert opinion
Proximity to wildlife	In wildlife, IDV has been detected only in feral swine but because of the wide range of hosts susceptible to infection, wild ruminants and other species from wildlife could play a role in transmission.	[5]
Proximity to water	Water areas can represent focus points where cattle from different farms can have close contact between each other and with wildlife, extensive breeding being the main breeding system for cattle and small ruminants in Togo.	Local expert opinion

**Table 2 viruses-12-00128-t002:** Number of samples collected by species and results after analysis.

Species	Nb. Sera Samples	Nb. IDV Seropositive Samples	Positive Sera (%) [Median HI Positive Titer; HI Titers Range]	Nb. Nasal Swabs	Nb. IDV Positive Swabs
Cattle	399	18	4.5 [20; 10–320]	10	0
Small ruminants	737	28	3.8 [40; 10–160]	840	0
Swine	80	0	0	346	0

Nb: number.

**Table 3 viruses-12-00128-t003:** Weights, risk relationships, and thresholds attributed by experts.

Risk Factor	Mean Weight	Risk Relationships *	Thresholds
Cattle density	0.38	Linear increasing	*a* = minimum raster layer value *b* = maximum raster layer value
Small ruminants density	0.08	Linear increasing	*a* = minimum raster layer value *b* = maximum raster layer value
Swine density	0.11	Linear increasing	*a* = minimum raster layer value *b* = maximum raster layer value
Proximity to water	0.01	Sigmoid decreasing	*a* = 2.5 km *b* = 5 km
Proximity to cattle market	0.24	Sigmoid decreasing	*a* = 5 km *b* = 10 km
Proximity to wildlife	0.02	Sigmoid decreasing	*a* = 2 km *b* = 4 km
Proximity to transhumance areas	0.16	Sigmoid decreasing	*a* = 0.5 km *b* = 2.5 km

* When risk relationships proposed by the experts were different, a consensus was derived giving more importance to local experts.

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
