# Peer review of "Risk Mapping of Influenza D Virus Occurrence in Ruminants and Swine in Togo Using a Spatial Multicriteria Decision Analysis Approach"

_viruses, 2020, doi:10.3390/v12020128_

Round 1

Reviewer 1 Report

In their manuscript, Fusade-Boyer and colleagues generated risk maps for the occurrence of influenza D virus in Togo using a spatial (Geographic Information System-based) multicriteria decision analysis (MCDA). The development of such risk maps for various infectious diseases is very important, especially in resource-poor countries, in order to optimize the surveillance system in these regions. For some more severe diseases, such as HPAIV, Rift valley fever and African swine fever such analysis has already been implemented for the assessment of the likelihood of their occurrence. Although influenza D virus has not been recognized as a pathogen causing severe disease in animals, serologic surveillance suggests that this virus readily infects ruminants and swine. Therefore, generating such risk maps for the IDV in African countries is also worthwhile.

The study was performed by a large international group of recognized experts in virology (IDV experts), epidemiology, veterinarian sciences and mathematic modelling – a successful combination for conducting such complex studies. An important evidence for the feasibility of the generated risk map is that its visual comparison with serological results showed that most of the sampled areas with no seropositivity coincided with low risk areas in the model. In contrast, IDV-seropositive samples came from mix of high and low risk areas. The maps generated by the authors can be used for targeted sampling of the ruminants and swine populations in Togo in future IDV surveillance programs.

The manuscript is well-written and contains all necessary details supporting the authors’ conclusions. I only have some minor comments to be addressed before the manuscript can be published.

Is the authors’ model universal? I.e. can it be implemented in other countries with similar risk factors? Lanes 54-55. What time period was used for sample collection? Lane 55. Please replace “pig” with “pigs” Lane 56. Were swab and sera samples matched? Lane 62. What was the starting dilution of the serum samples for HI tests?

Author Response

We thank the Reviewers for their valuable comments and the opportunity to improve our manuscript before publication in Viruses. Below is a point by point answer to Reviewers’ questions and comments.

 Reviewer: 1 

In their manuscript, Fusade-Boyer and colleagues generated risk maps for the occurrence of influenza D virus in Togo using a spatial (Geographic Information System-based) multicriteria decision analysis (MCDA). The development of such risk maps for various infectious diseases is very important, especially in resource-poor countries, in order to optimize the surveillance system in these regions. For some more severe diseases, such as HPAIV, Rift valley fever and African swine fever such analysis has already been implemented for the assessment of the likelihood of their occurrence. Although influenza D virus has not been recognized as a pathogen causing severe disease in animals, serologic surveillance suggests that this virus readily infects ruminants and swine. Therefore, generating such risk maps for the IDV in African countries is also worthwhile.The study was performed by a large international group of recognized experts in virology (IDV experts), epidemiology, veterinarian sciences and mathematic modelling – a successful combination for conducting such complex studies. An important evidence for the feasibility of the generated risk map is that its visual comparison with serological results showed that most of the sampled areas with no seropositivity coincided with low risk areas in the model. In contrast, IDV-seropositive samples came from mix of high and low risk areas. The maps generated by the authors can be used for targeted sampling of the ruminants and swine populations in Togo in future IDV surveillance programs.

The manuscript is well-written and contains all necessary details supporting the authors’ conclusions. I only have some minor comments to be addressed before the manuscript can be published.

Thank you very much for your interest in this work.  

Is the authors’ model universal? I.e. can it be implemented in other countries with similar risk factors?  

Our model is the first to map the risk of influenza D virus and it was developed in a small country (Togo). We therefore do not think this model is universal. It could however be implemented with similar risk factors in countries with the same kind of livestock system (Benin for example), but it would still need to be validated. Lanes 54-55. What time period was used for sample collection?  We have added details regarding the collection period. It reads as follows (line 55-56): 

“Through the national disease surveillance program, 2,412 samples were collected from cattle, small ruminants and pigs, in markets and farms in 22 different locations in Togo between 2017 and 2019.” Details on the transhumance/non transhumance period were also added (see reply to Reviewer 2).

Lane 55. Please replace “pig” with “pigs” Lane 56.  

It is now corrected, thank you (line 55)  

Lane 56. Were swab and sera samples matched?  

No, sera and swabs were not necessarily matched. We have now added a line to clarify this point in the manuscript. It reads as follows (line 56-57):

 “These samples included 1,196 nasal swabs and 1,216 sera (not necessarily paired samples from the same individuals).”   

Lane 62. What was the starting dilution of the serum samples for HI tests? 

The started dilution of serum samples for HI assay is 1:10. We have now clarified this point (line 61):

“All sera were treated with receptor destroying enzyme (RDE, Seika) following the manufacturer’s instructions, diluted 10 folds and hemadsorbed on packed chicken red blood cells.”  

Reviewer 2 Report

Submitted MS from Fusade-Boyer employs multicriteria decision analysis (MCDA) analysis to draw influenza D virus risk maps for Togo with a special focus on swine and cattle. I believe it is an important contribution, especially with respect to Togo agricultural/government authorities, who could use the presented work to address and decrease risk. The authors identified multiple risk factors and assigned them with relative value.

By overlaying data over the geographical map of Togo authors created pams that identify the highest risk areas and reflect the temporal dynamics of the relative risk.

In addition, the authors collected swabs and sera samples from swine and cattle host to test acute infection as well as seroprevalence of (attachment inhibitory) functional antibodies and addressed geographical seroprevalence distribution with risk maps created. Paper is well written easy to read.

Comments:

MS has more local importance unless the goal is to present a platform to make these MCDA maps for any region.

If that is the case they should create a risk map for someplace where, comprehend seroprevalence maps are available, because that would be the validation of the process. Then they could present the map for Togo as a predictive tool. That would give the paper broad importance.

While seroprevalence correlates so and so with map's high-risk areas, there is no seroprevalence increase in cattle transhumance areas. Is it because of samples availability? 

Authors should include a description of samples collection (dates, areas, with respect to transhumance, that is obviously important)

As hemagglutination inhibition targets attachment inhibition antibodies only, this is not the only way to address the question of anti-IDV antibody prevalence. The authors should perform ELISA assay as that would provide a more robust approach.

I assume that cattle are not vaccinated, but authors could include a line or two to clarify this. If that is the case all the abs detected come from natural infection.

In addition, as HA-NA is one protein, esterase inhibition titers along with HI titers should be tested and reported, perhaps as a table.

Table2 is only presenting positivity (HI >10), please present the numbers 

At least two antigenically distinct strains were described already in 2014 (DOI: 10.1128/JVI.02718-14), authors need to include in addition to D/bovine/France/5920/2014 another antigenic variant/s of IDV or otherwise justify the employment of only one antigenic variant (some reference showing that only this antigenic variant was is prevalent in Togo)

the legend in fig.3 protrudes to the map

Overall, I would recommend medium changes to MS.

Author Response

Reviewer: 2

Submitted MS from Fusade-Boyer employs multicriteria decision analysis (MCDA) analysis to draw influenza D virus risk maps for Togo with a special focus on swine and cattle. I believe it is an important contribution, especially with respect to Togo agricultural/government authorities, who could use the presented work to address and decrease risk. The authors identified multiple risk factors and assigned them with relative value.

By overlaying data over the geographical map of Togo authors created pams that identify the highest risk areas and reflect the temporal dynamics of the relative risk.

In addition, the authors collected swabs and sera samples from swine and cattle host to test acute infection as well as seroprevalence of (attachment inhibitory) functional antibodies and addressed geographical seroprevalence distribution with risk maps created. Paper is well written easy to read.

Thank you very much for your comments.

Comments:

MS has more local importance unless the goal is to present a platform to make these MCDA maps for any region.

If that is the case they should create a risk map for someplace where, comprehend seroprevalence maps are available, because that would be the validation of the process. Then they could present the map for Togo as a predictive tool. That would give the paper broad importance.

The main objective of our study was indeed local. Because of very low titers, low seropositivity rates and the fact that no sample was found positive by PCR in Togo, we hope that this work will allow for increasing chances to detect IDV in this country. Nevertheless, the maps can indeed not be fully validated in Togo and drawing maps for others countries with reliable epidemiological data would be a good way to build a predictive tool that we could apply in Togo. No data is unfortunately available in another Western African country as of now to our knowledge to be able to validate the model in the proper context. Countries with good seroprevalence data are Western countries, where livestock systems are very different from those of Africa. Even if the maps were correctly validated for Western countries, it would not necessarily mean that the model may be a predictive tool in African countries, with probably differences in risk factors weights. Increasing the number of samples to validate the map is another possibility but not possible unfortunately with our current surveillance project.  

While seroprevalence correlates so and so with map's high-risk areas, there is no seroprevalence increase in cattle transhumance areas. Is it because of samples availability? 

Yes indeed, only one transhumance area was sampled during the transhumance period and found positive but all the others were not sampled so far, whether during or outside the transhumance period.

Authors should include a description of samples collection (dates, areas, with respect to transhumance, that is obviously important)

As very few samples were collected in transhumance areas, it was not possible to consider this factor to analyze serological results. The same is true for the transhumance period. Indeed, among the 723 sera collected during the transhumance period there were all the cattle sera (399 sera) (0 cattle sera were collected outside the transhumance period), which is a significant bias of our sampling here.

We have added details in the results (line 150-151) and in the discussion sections (lines 268-273) to clarify this point. It reads as follows:

Result section:

“Surprisingly, 43 positive samples were collected during the transhumance period and only 3 outside the transhumance period.”

Discussion section:

“Interestingly, in the present study, 43 positive sera among 723 were collected during the transhumance period and only 3 positive sera among 493 were collected outside the transhumance period. These results should however be considered with caution. Indeed, all the 399 cattle sera were collected during the transhumance period, which represents a significant bias. Another significant bias is the collection areas, since some places were sampled only during transhumance period and some others only outside the transhumance period.”

As hemagglutination inhibition targets attachment inhibition antibodies only, this is not the only way to address the question of anti-IDV antibody prevalence. The authors should perform ELISA assay as that would provide a more robust approach.

We agree with the Reviewer, according to Moreno et al. 2019, the ELISA is more sensitive than HI test. It would be indeed more accurate to use the ELISA. Unfortunately, we do not have this tool yet in the lab but we it should be implemented in the coming months. We have added a sentence in the discussion to clarify this point. It reads as follows (lines 252-255):

“Seropositivity rates we calculated may also have been underestimated because HI assays are less sensitive than ELISAs which should be preferred for further serological analyses [31], even if previous IDV seroprevalence in Africa were calculated from HI assays data as well [2].”

I assume that cattle are not vaccinated, but authors could include a line or two to clarify this. If that is the case all the abs detected come from natural infection.

Yes indeed, cattle are not vaccinated against IDV and we have now added a sentence to clarify this point. It reads as follows (lines 256):

“As no IDV vaccination is in place, all the positive sera came from natural infections.”

In addition, as HA-NA is one protein, esterase inhibition titers along with HI titers should be tested and reported, perhaps as a table.

We agree with the Reviewer but esterase inhibition tests are not common and were not carried out on the sera tested here. We however only needed to detect anti-IDV antibodies and we believe that antibodies against the hemagglutinin part of HEF are sufficient and predominant.

Table2 is only presenting positivity (HI >10), please present the numbers 

We thank the Reviewer for the suggestion and now included a median HI titer and a range of HI titers for the positive sera in Table 2.

At least two antigenically distinct strains were described already in 2014 (DOI: 10.1128/JVI.02718-14), authors need to include in addition to D/bovine/France/5920/2014 another antigenic variant/s of IDV or otherwise justify the employment of only one antigenic variant (some reference showing that only this antigenic variant was is prevalent in Togo)

Yes indeed, two antigenic variants of IDV circulate in North America and Europe. In the previous study on IDV in Africa (Salem et al. 2017) two strains were used for HI assays: D/bovine/France/5920/2014 (D/swine/Oklahoma-like lineage) and D/bovine/Nebraska/9-5/2012 (D/bovine/660-like lineage), and no significant differences were found between the 2 antigens. We therefore only used D/bovine/France/5920/2014 here. We have now clarified this point adding a sentence in the Material and Methods section of the manuscript, it reads as follows (lines 63-66):

“Based on our previous study carried out on African sera, a single antigen was used in HI tests although two distinct genetic and antigenic lineages of IDV were reported so far in North America and Europe (D/swine/OK and D/bovine/660) [7] (no significant differences were observed between the 2 antigens in HI assays on African sera [2]).”

the legend in fig.3 protrudes to the map

This was now corrected.

Overall, I would recommend medium changes to MS.